# Doxorubicin Loading into Milk and Mesenchymal Stem Cells’ Extracellular Vesicles as Drug Delivery Vehicles

**DOI:** 10.3390/pharmaceutics15030718

**Published:** 2023-02-21

**Authors:** Anindya Mukhopadhya, Dimitrios Tsiapalis, Niamh McNamee, Brian Talbot, Lorraine O’Driscoll

**Affiliations:** 1School of Pharmacy and Pharmaceutical Sciences, Panoz Institute, Trinity College Dublin, D02 PN40 Dublin, Ireland; 2Trinity Biomedical Sciences Institute, Trinity College Dublin, D02 R590 Dublin, Ireland; 3Trinity St. James’s Cancer Institute, D08 W9RT Dublin, Ireland

**Keywords:** drug-loading, milk, mesenchymal/stromal stem cells, extracellular vesicles, doxorubicin, electroporation, sonication, HPLC, imaging flow cytometry

## Abstract

Extracellular vesicles (EVs) have great potential as drug delivery vehicles. While mesenchymal/stromal stem cell (MSC) conditioned medium (CM) and milk are potentially safe and scalable sources of EVs for this purpose, the suitability of MSC EVs and milk EVs as drug delivery vehicles has never been compared and so was the objective of this study. Here EVs were separated from MSCs’ CM and from milk and were characterised by nanoparticle tracking analysis, transmission electron microscopy, total protein quantification, and immunoblotting. An anti-cancer chemotherapeutic drug, doxorubicin (Dox), was then loaded into the EVs by one of three methods: by passive loading or by active loading by either electroporation or sonication. Dox-loaded EVs were analysed by fluorescence spectrophotometer, high-performance liquid chromatography (HPLC), and imaging flow cytometer (IFCM). Our study showed that EVs were successfully separated from the milk and MSC CM, with significantly (*p* < 0.001) higher yields of milk EVs/mL starting material compared to MSC EVs/mL of starting material. Using a fixed amount of EVs for each comparison, electroporation achieved significantly more Dox loading when compared to passive loading (*p* < 0.01). Indeed, of 250 µg of Dox made available for loading, electroporation resulted in 90.1 ± 12 µg of Dox loading into MSC EVs and 68.0 ± 10 µg of Dox loading into milk EVs, as analysed by HPLC. Interestingly, compared to the passive loading and electroporation approach, after sonication significantly fewer CD9+ EVs/mL (*p* < 0.001) and CD63+ EVs/mL (*p* < 0.001) existed, as determined by IFCM. This observation indicates that sonication, in particular, may have detrimental effects on EVs. In conclusion, EVs can be successfully separated from both MSC CM and milk, with milk being a particularly rich source. Of the three methods tested, electroporation appears to be superior for achieving maximum drug loading while not causing damage to EV surface proteins.

## 1. Introduction

There is a need for the development and head-to-head comparison of sources of potentially reliable, cost-effective, scalable, and easy-to-produce vehicles for the delivery of therapeutics, in addition to options such as liposomes [1,2]. The importance of this is exemplified by the fact that liposomes have been associated with adverse influences such as immunogenicity and uncontrolled toxicity, in several liposome-based drug delivery studies [3,4,5]. 

Extracellular vesicles (EVs) from various cells and biological fluids (i.e., plasma and milk) have been proposed as an ideal, nature-derived, drug-delivery vehicle [6,7,8]. The term EVs refers to membrane-surrounded vesicles released from cells that may be engaged in cell-to-cell communication [9]. Desirable characteristics of EVs as drug delivery carriers include their intrinsic ability to cross biological barriers including the blood-brain barrier, non-immunogenicity, and non-toxicity properties [6,10]. Moreover, EVs have been reported to demonstrate superior targeting capabilities to liposomes [8]. A proof-of-principle study from our group showed that EVs can be harnessed as anti-cancer therapeutic delivery vehicles [11]. Of course, due to the inherent other unwanted cargo in cancer cell EVs, these would not be suitable to bring forward toward clinical utility.

Two potential sources of safe EVs as vehicles for pharmaceutical agents are EVs from mesenchymal/stromal stem cells (MSCs) [12,13,14,15,16] and EVs from milk [17,18,19]. In relation to MSC EVs, a protocol was deemed suitable for separating EVs from the conditioned medium (CM) of MSC for subsequent successful administration, as therapy in the clinic in their natural (non-loaded) form has been reported [20]. For milk EVs, we recently developed an effective protocol to separate relatively pure EVs from milk, which involved the removal of caseins, which are highly abundant in milk and overlap in size with EVs [21,22]. 

Furthermore, both active and passive drug-loading approaches have been established to load therapeutic molecules into EVs, as reviewed by Mehryab et al. [7]. However, the is a lack of direct comparative studies of such loading approaches for EVs from different sources as delivery vehicles to help select the optimal processes to bring forward. 

Hence, the overall aim of this study (Figure 1) was to collect and characterise MSC EVs and milk EVs and compare the success of their using a passive loading approach and two active loading approaches, i.e., electroporation versus sonication loading. The anti-cancer drug doxorubicin (Dox) was chosen as a cargo of interest. In this study, we focused on investigating the loading of Dox into EVs as well as evaluating the effects of loading approaches on key biological aspects of EVs. 

## 2. Materials and Methods

### 2.1. EVs Separation Methods

MSC EVs separation: Bone marrow MSCs (Stemcell Technologies, Cambridge, UK) were cultured in low glucose Dulbecco’s modified Eagle medium (DMEM) (Sigma-Aldrich, Wicklow, Ireland) supplemented with 1% L-glutamine (Sigma-Aldrich, Wicklow, Ireland), 1% penicillin-streptomycin (Sigma-Aldrich, Wicklow, Ireland), 5 IU/mL heparin (Sigma-Aldrich, Wicklow, Ireland), and 10% human platelet lysate (hPL; Stemcell Technologies, Cambridge, UK) and maintained at 37 °C in a humidified atmosphere with 5% CO_2_. The medium was changed every 3–4 days. At passage 6 and approximately 70% confluency, MSCs were washed with phosphate-buffered saline (PBS, Sigma-Aldrich, Wicklow, Ireland) and changed to a medium containing EVs-depleted FBS. After 48 h, a conditioned medium (CM) was collected. Any floating cells that remained in the medium were eliminated from the CM (100 mL) by centrifugation at 2000× *g* for 10 min. To remove cellular debris, supernatants were centrifuged at 10,000× *g* for 45 min at 4 °C in a Sorvall™ ST 8 small benchtop centrifuge using an HIGHConic™ III fixed angle rotor (Thermo Fisher Scientific, Dublin, Ireland), followed by filtration using 0.2-µm pore filters (Fisher Scientific, Dublin, Ireland). Next, EVs were separated using the polyethylene glycol 6000 (PEG) precipitation protocol, as previously described [20], with a modified centrifugation step, i.e., ultracentrifugation was performed at 100,000× *g* for 91 min using a Ti70 rotor in Optima XPN-100 Ultracentrifuge (Beckman Coulter, Brea, CA, USA) at 4 °C. MSC-EVs and preparations were resuspended in 500 µL of 10 mM Hepes/0.9% NaCl (Thermo Fisher Scientific, Dublin, Ireland). The EVs samples were stored at −80 °C until further use.

Milk EVs separation: commercially available bovine skim milk samples were used for milk EV separation using a protocol that we developed recently [21,22]. Briefly, caseins, the major EVs contaminant proteins present in milk were removed from the milk samples by isoelectric precipitation. For this, the pH was dropped to 4.6 using 6N HCl and the samples were then centrifuged at 8000× *g* at 20 °C for 30 min. Clear whey (2.33 mL) was used for density gradient ultracentrifugation following the bottom-up technique using Optiprep iodixanol solution (60%). Ultracentrifugation was performed using an SW Type 32.1 Ti rotor in Optima XPN-100 Ultracentrifuge (Beckman Coulter, Brea, CA, USA) at 186,000× *g* for 18 h at 4 °C. Pooled fractions (1.05–1.20 g/mL) were washed twice in sterile PBS, followed by 120,000× *g* for 2 h at 4 °C. The pellet was resuspended in 500 µL sterile PBS and EVs samples were stored at −80 °C until further use.

### 2.2. Fundamental Characterisation of EVs

For the fundamental characterisation of EVs separated from milk and MSCs, Bradford assay, nanoparticle tracking analysis (NTA), transmission electron microscopy (TEM) and immunoblotting were performed.

#### 2.2.1. Bradford Assay for Protein Quantification

Bradford assay was performed to determine the amount of total protein in samples using Bio-Rad Protein Assay Dye Reagent Concentrate (Bio-Rad Laboratories, Watford, UK, Cat #500-0006). 

#### 2.2.2. Nanoparticle Tracking Analysis

The particle concentration and sizes were estimated using NanoSight NS300 (Malvern Instruments Ltd., Malvern, UK). Milk EVs and MSC EVs samples were automatically injected into the NTA system under constant flow conditions (flow rate = 50); videos of the particles in motion were recorded and analysed using NTA 3.1.54 software.

#### 2.2.3. Transmission Electron Microscopy

Samples were prepared for transmission electron microscopy (TEM) analysis following the previously published protocol [23]. Specifically, 5 μL of the sample was suspended in 5 μL of 0.22 μm-filtered PBS and placed onto carbon-coated grids (Ted-Pella B 300M, Mason Technology Ltd., Dublin, Ireland Cat. #: 01813-F) for 10 min at room temperature before fixing with 4% glutaraldehyde (Sigma-Aldrich, Wicklow, Ireland) and contrasting with 2% phosphotungstic acid (Sigma-Aldrich, Wicklow, Ireland). The grids were examined at 100 kV using a JEOL JEM-2100 TEM (JOEL, Peabody, MA, USA), as we previously described [24]. 

#### 2.2.4. Immunoblotting

Immunoblotting was performed as previously described [24]. MSC EVs and milk EVs were lysed using SDS lysis buffer (250 nM Tris-HCL, pH 7.4; 2.5% SDS); cell lysates from MSCs and Hs578Ts(i)8 were used as a positive control for the proteins being analysed, respectively. A quantity of 20 μg of EVs or cell lysates was resolved on 10% SDS gels (Bio-Rad Laboratories, Watford, UK) and the protein was transferred onto PVDF membranes (Bio-Rad Laboratories, Watford, UK, Cat. #1620177). Blots were blocked with 5% (*w*/*v*) BSA in PBS containing 0.1% Tween-20 and incubated overnight at 4 °C with primary antibodies to TSG101 (1:1000; Abcam, Cambridge, UK, Cat. # ab30871), CD63 (1:500; Abcam, Cat. #: ab68418), actinin-4 (1:1000, Abcam, Cat. #: ab108198), calnexin (1:1000, Abcam, Cat. #: ab133615), syntenin-1 (1:1000, Abcam, Cat. #: ab133267), and CD9 (1:1000, Abcam, Cat. #: ab92726). Secondary antibodies used were anti-mouse (1:1000 in 5% BSA/PBS-T, Cell Signalling, Cat. #: 7076) or anti-rabbit (1:1000 in 5% BSA/PBS-T, Cell Signalling, Cat. #: 7-74). SuperSignal West Femto Chemiluminescent Substrate Kit (Fisher Scientific, Cat. # 11859290) was used for detection, imaging was performed using an automated Chemidoc exposure system (Bio-Rad Laboratories, Watford, UK).

### 2.3. Doxorubicin Loading into EVs

#### 2.3.1. Passive Loading

A total of 250 µg of doxorubicin hydrochloride (Dox) (Sigma-Aldrich, Wicklow, Ireland) was added to 125 µg of MSC EVs or milk EVs, and the Dox-EVs mix was incubated on a thermo-mixer at 37 °C for 2 h.

#### 2.3.2. Active Loading

Two different active loading approaches were performed, as described below:(i)Electroporation: The electroporation of MSC EVs and milk EVs to load Dox was performed using a Neon™ Transfection System (Thermo Fisher Scientific, Dublin, Ireland). A total of 3 mL of Electrolytic buffer E was added to Neon™ tube and fixed to the tube stand. A total of 250 µg of Dox was mixed with 125 µg of MSC EVs or milk EVs and placed on ice. A 100 µL Neon™ Tip was attached to the Neon™ pipette and 100 µL of EV-Dox mix was pipetted up, the pipette was fixed to the Neon™ pipette station and electroporation protocol was performed using the program set on the Neon™ device at 1500 volts. The electroporated sample was collected in a fresh tube, and equal volume of Resuspension buffer R was added and incubated at 37 °C for 1 h for recovery of the EVs membrane.(ii)Sonication: The sonication procedure was performed using a microson ultrasonic cell disruptor with a 0.25 tip (Misonix Inc., New York, NY, USA) with the following settings: 20% amplitude, and 6 cycles of 30 s on/off for 4 min with a 2 min cooling period between each cycle. The sonicated samples were incubated at 37 °C for 1 h to allow for recovery of the EVs membrane.

Following the loading step, unbound Dox was removed by washing the loaded EVs using a 300 kDa filter (Nanosep, Pall Biotech, Crosshaven, Ireland).

### 2.4. Fluorescence Analysis of Doxorubicin

The percentage of Dox made available that was subsequently successfully loaded in EVs was analysed by measuring the intrinsic fluorescence of doxorubicin using a fluorescence plate reader (Spectra Max Gemini System) with excitation and emission wavelengths of 485 and 535 nm, respectively. A standard curve of doxorubicin was used to interpolate unknown quantities of Dox.

### 2.5. High-Performance Liquid Chromatography Analysis

To evaluate Dox loading, high-performance liquid chromatography (HPLC) analysis was performed using the Waters Alliance HPLC system (Waters Corporation, Milford, MA, USA). An amount of 20 µL of lysed or non-lysed Dox-loaded milk EVs and MSC EVs samples were injected and run on a C18 column (Thermo Fisher Scientific, Dublin, Ireland) using a mobile phase of acetonitrile: water (pH 3) at a ratio of 70:30 (*v*/*v*) at a flow rate of 1 mL/min at 30 °C. Absorbance was measured at 227 nm to monitor the elution of Dox.

### 2.6. Imaging Flow Cytometry Analysis

The imaging flow cytometry (IFCM) analysis of Dox loaded was performed as previously described [21]. The specific antibodies used in this study were anti-CD63 conjugated with Fluorescein isothiocyanate (FITC) CD63-FITC, 1:150, Cat. #:353006, Biolegend, San Diego, CA, USA); CD9-PE (1:1500, Cat. #: 312106, Biolegend, San Diego, CA, USA); and CD81-PE-Cy7 (1:150, Cat. #: 349,512 Biolegend, San Diego, CA, USA). EVs were incubated with the antibodies for 45 min at RT in the dark and washed using a 300 kDa filter (Nanosep, Cat. #: 516-8531). Samples were resuspended in 50 µL IFCM buffer and acquired within 2 h on the ImageStream X Mk II imaging flow cytometer (Amnis/Luminex, Seattle, USA) at 60x magnification and low flow rate and EVs-free IFCM buffer, unstained EVs, single-stained controls, and fluorescence minus one (FMO) controls were run in parallel. EVs were gated as SSC-low vs. fluorescence, then as non-detectable brightfield (fluorescence vs. Raw Max Pixel Brightfield channel); gated EVs were confirmed in the IDEAS Image Gallery. Data analysis was performed using IDEAS software v6.2 (Amnis/Luminex, Seattle, WA, USA).

### 2.7. Data Analysis

For all experiments, data are presented as mean ± SEM. Tests for significant differences between the groups were performed using a t test or one-way ANOVA with multiple comparisons using Graph Pad Prism 9 software. A value of *p* < 0.05 was considered statistically significant.

## 3. Results and Discussion

### 3.1. MSC EVs and Milk EVs Separation and Characterisation

EVs were successfully separated from MSC CM and milk and characterised by NTA, TEM, and immunoblotting analysis. As presented in Appendix A, the complete removal of contaminant milk protein, casein, was confirmed by comparing the same with untreated milk EV samples by polyacrylamide gel electrophoresis. The size and concentration of EVs particles obtained from MSC and milk are presented in Figure 2a. As estimated by NTA, the sizes of MSC EVs/particles (mean: 114.6 ± 2.7 nm and mode: 92.8 ± 10.8 nm) were smaller compared to milk EVs particles (mean: 169.3 ± 2.8 nm and mode: 146.8 ± 2.8 nm). The yield of EVs from MSC CM (6.22 × 10^8^ ± 4.46 × 10^7^ particles/mL of starting material) was significantly lower (*p* < 0.001) than milk (2.06 × 10^10^ ± 1.04 × 10^9^ particles/mL of starting material). Generally, the use of mammalian cell cultures including MSCs to scale up EVs yield remains challenging for many reasons including specialised equipment, costs, maintaining the MSC phenotype, etc. Based on this comparison, the alternative use of milk as an EVs source would seem to be advantageous due to milk’s easy accessibility, cost-effectiveness, and scalability potential. Of course, not only the source of EVs needs to be considered, but also the EVs’ collection method plays a role in the yield and purity of EVs. One EVs collection method is not optimal for all biofluids, each has its own complex matrix. Therefore, although MSC CM resulted in fewer EVs/mL in this specific comparison, given that MSC EVs have some reported endogenous inflammation supressing/immune-modulatory capabilities, in instances where these additional beneficial characteristics might be considered useful then MSC seeding densities could potentially be scaled up to produce greater EVs yield/mL of CM. We would suggest, therefore, that based on yield per mL of starting material, MSC EVs should not be ruled out. Furthermore, TEM analysis of EVs samples showed that intact EVs-like structures were observed in both MSC EVs and milk EVs samples (Figure 2b), suggesting that, based on these fundamental characteristics, both MSC EVs and milk EVs were suitable to process further in their comparison as potential delivery vehicles.

As an additional characterisation step, in keeping with MISEV2018 guidelines, four proteins considered to be positive markers for EV populations and two proteins considered to be negative markers for EV populations were analysed by immunoblotting. Cell lysate from the Hs578Ts(i)8 human cell line was included in all gels as a positive control [21]. As presented in Figure 2c, both MSC EVs and milk EVs were positive for TSG101, CD63, syntenin-1, and CD9, whereas no signals were obtained for the negative markers actinin-4 and calnexin.

Therefore, the successful separation of intact EVs from MSC and milk was confirmed using the characterisation methodologies.

### 3.2. Comparison of Doxorubicin Retention and Protein Analysis Post-Drug Loading

Having successfully separated EVs from MSC CM and milk, 250 µg of the chemotherapeutic drug, Dox, was added to EVs (125 µg) from each source for loading by either passive loading or by either method of active loading (Figure 1). The apparent Dox loading achieved, i.e., the percentage of Dox that was made available with a fixed amount of EVs and that remained after removing extraneous Dox by rounds of washing is presented in Figure 3a. The electroporation approach apparently resulted in the most successful Dox loading, when compared to passive loading (*p* < 0.01) for both MSC EVs (achieving 86.4%) and milk EVs (achieving 80%). Passive loading of MSC EVs (47.2% achieved) and milk EVs (38% achieved) was also associated with significantly (*p* < 0.05) lower Dox loading capacity when compared to sonication (MSC EVs (75% achieved) and milk EVs (64% achieved) (Figure 3a). Using total protein quantities as a surrogate for the EVs post-load of those from the two sources and with each of the three methods, total protein quantities did not differ significantly from the amounts made available to the drug (Figure 3b). This indicated no substantial overall loss of EVs with any loading method.

Interestingly, in previous studies where Dox was loaded into milk EVs, 21% of encapsulation of the available drug was achieved by passive loading (Li et al. [25])and 13.4% loading via a conjugation method (Zhang et al. [26]). Crucially, neither Li et al. [25] nor Zhang et al. [26] reported the removal of the predominant milk proteins before the collection of EVs. This is particularly important as casein aggregates are abundant in the milk [27] and are of a size range overlapping with EVs [21]. Hence, their presence may lead to some Dox being caught up in these protein aggregates and so may explain the lower amount of Dox loading into EVs in those studies. In contrast, our optimised EVs separation protocol ensured maximum removal of casein aggregates [22], resulting in relatively high quantities of pure milk EVs available for loading. In the case of Dox loading to MSC EVs by electroporation, Gomari et al. [28] and Bagheri et al. [29] reported only 47% and 13% success, respectively. In contrast, using with a similar electroporation approach, we achieved 86.4% success. It has been reported that, in general, low purity and low yields of EVs are the major limitations for use of EVs in drug delivery [30,31]; therefore, our maximising purity by removing caseins seem to be a positive step in this regard.

Overall, when the results of Dox loading success achieved in the context of total protein remaining (as a surrogate of EVs quantities) are considered at this stage in the comparative study, it seemed that electroporation is the most favourable loading approach for both MSC EVs and milk EVs.

### 3.3. Analysis of Dox Quantity Loaded in MSC and Milk Evs by HPLC

It could be proposed that, despite the washes to remove extraneous Dox after completing the loading step, some Dox detected by fluorescence may be associated with Evs but not in the Evs as cargo. HPLC is reported to be an accurate, reproducible, and high throughput method for the quantification of small chemotherapeutic drug molecules such as Dox, paclitaxel, or curcumin [7,32]. Thus, to further investigate the amount of Dox loaded into the Evs, paired aliquots of loaded Evs, both non-lysed (so surface Dox detected) versus lysed (ruptured, so Dox cargo also detected) were analysed by HPLC. Serially diluted Dox was analysed by HPLC to prepare a standard curve (Appendix A). The representative Dox peaks are presented in Appendix A. With passive-loaded Evs, no differences were observed between non-lysed and lysed MSC Evs or milk Evs (Figure 4). This indicates that most Dox signals detected in the passive loading protocol (as presented in Figure 3a) were unlikely due to the incorporation of Dox into Evs as cargo, but rather due to a simpler association/conjugation of Dox with Evs.

With respect to either the electroporation or sonication loading approaches, no peak was detected for Dox with the non-lysed samples. Rather, Dox was only detected when the Evs were lysed. This indicates that the Dox has been entrapped within the Evs as cargo. Specifically, following the electroporation method, 90.1 ± 12 µg and 68.0 ± 10 µg of Dox were detected for lysed Dox-loaded MSC Evs and milk Evs, respectively. In contrast, following the sonication method 38.3 ± 13 µg and 29.5 ± 9 µg of Dox were detected for lysed Dox-loaded MSC Evs and milk Evs, respectively. Overall, this data suggests that the electroporation process is associated with greater loading of Dox into MSC Evs and milk Evs as cargo, when compared to the passive loading and sonication approaches.

### 3.4. Effect of Passive and Active Loading Approaches on EVs Surface Markers

Imaging flow cytometry (IFCM) is a flow cytometry-based sophisticated method suitable for characterising EVs. This method is also useful to enumerate EVs specific surface markers and is recently reported as a robust method of EV analysis [32], even though none of the previously reported studies have analysed drug-loaded EVs with this robust technique for multi-parametric EVs analysis. Therefore, the enumeration of detectable EVs (pre- and post-loading) and amounts of three tetraspanins proteins are considered to be markers of EVs, i.e., CD9, CD63, and CD81 were analysed in order to investigate any negative effects of the various loading approaches on representative surface proteins.

The effect of loading on the number of objects/mL of MSC EVs and milk EVs are presented in Figure 5a,b, respectively. For both MSC EV and milk EV samples, there were significantly lower objects/mL detected in loaded samples compared to the non-loaded EV samples (*p* < 0.00001). Interestingly, the detected EV-like objects/mL post sonication was significantly lower compared to both passive loading (*p* < 0.05) and electroporation (*p* < 0.01) approaches, observed in both MSC EVs and milk EVs samples. There were no significant differences observed between the passive loading and electroporation approach in both MSC EVs and milk EVs samples. Hence, this observation indicates that sonication is a harsh approach that leads to the loss of intact EVs. Sonication has been known to be a harsh technique that can cause irreversible disruption to the cellular membrane [33], this may explain the lower number of detectable EVs following sonication (Figure 5a,b). Additionally, Zubair Ahmed et al. [34] reported that ‘low-power sonication’ altered MSC EVs membrane integrity. 

Further analysis of representative EV surface markers was performed on the loaded MSC EVs and milk EVs.

Overall, similar quantities of surface markers, in terms of positive objects/mL, were observed in MSC EVs compared to milk EVs; CD9 was the most abundant, followed by CD63 and CD81. It is of note that when IFCM data was compared to immunoblotting data (Figure 2c), some differences were evident, especially in the abundance of CD9 and CD63. It must be considered, however, that immunoblotting is a semi-quantitative way of assessing these proteins in lysed samples of the overall pool of EVs, while IFCM is quantitative and is evaluating surface proteins on individual intact EVs. 

As presented in Figure 5c, for MSC EVs sonication was associated with significantly (*p* < 0.0001) lower numbers of CD63+ and CD9+ objects/EVs, compared to passive loading and electroporation. Sonicated loaded samples also had significantly (*p* < 0.05) fewer CD81+ objects when compared to those loaded by electroporation. With milk EVs, similar observations rang through (Figure 5b) as observed with MSC EVs. 

This indicates that, at least with the electroporation setting that we optimised for this study, the EVs surface protein were conserved (apparently as gentle on them as passive loading), while sonication produced detrimental effects on the EVs, i.e., lower number of EVs and proteins, particularly CD63 and CD9. This further substantiates our observation that electroporation is a preferable drug loading approach over sonication and passive loading, while no major differences were detected between milk EVs and MSC EVs in this regard.

Interestingly, a literature search indicated that a clear consensus on the ‘best drug loading approach’ has not been reached among EVs researchers and comparative studies focusing on loading approaches and use of different EVs sources have been lacking [8,35,36]. 

## 4. Conclusions

Our study showed that electroporation is a preferable drug loading approach, when compared to both passive loading and sonication. With electroporation, greatest success was reached in loading Dox into the EVs and minimal negative effects on surface proteins were detected, while sonication seemed to be detrimental to these proteins. When comparing MSC EVs to milk EVs, no substantial differences in Dox loading capacity were observed between electroporated MSC EVs and milk EVs, indicating that both are equally suitable. Thus, the decision to select one over the other may depend on the further future experiments that will take into consideration that MSC EVs have been shown to have some natural anti-inflammatory/immune modulating characteristics that may be of benefit to a given application in addition to its cargo/loaded therapeutic content. On the other hand, milk is cheap, easily accessible, easily scalable, and very rich source of EVs and so may be more favourable in other circumstances, e.g., for oral delivery.

## Figures and Tables

**Figure 1 pharmaceutics-15-00718-f001:**
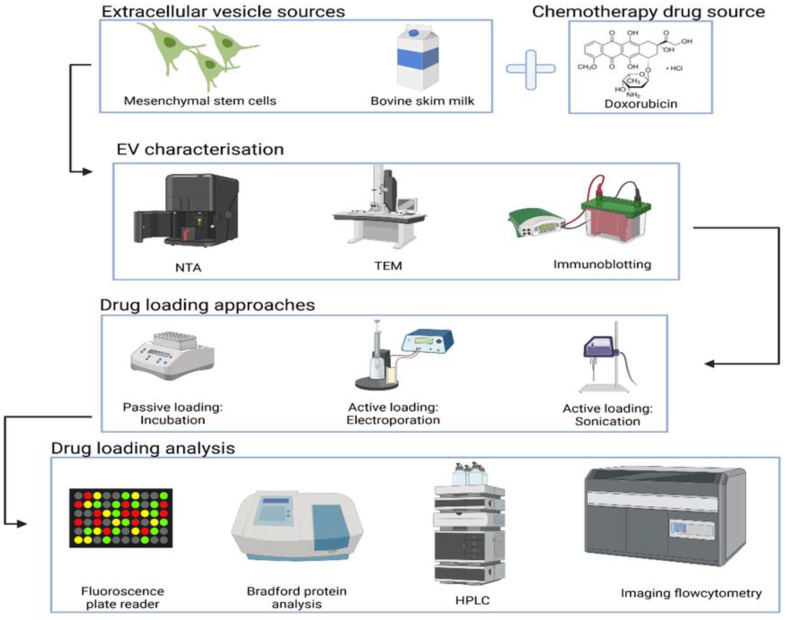
Illustrative summary of this comparative study. EVs were separated from MSCs and milk, characterised, and loaded with Dox by either passive loading (incubation) or active loading (electroporation or sonication). Both Dox loading successes and potential negative effects of loading on EVs (assessed for total protein as a surrogate and for established EVs surface proteins) were evaluated.

**Figure 2 pharmaceutics-15-00718-f002:**
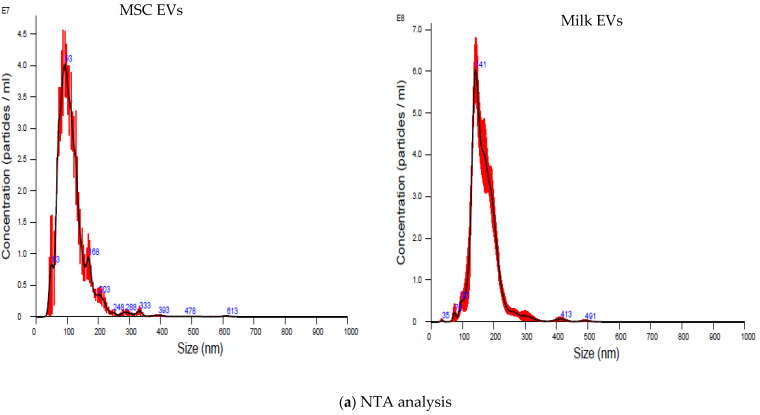
Fundamental characterisation of MSC EVs and milk EVs. MSC EVs and milk EVs (**a**) size distribution estimates obtained by NTA analysis and generated using NTA 3.1.54 software. (**b**) Representative TEM images (scale bar = 100 nm). (**c**) Immunoblots (using 20 μg of EVs lysates or cell lysates) for TSG101, CD63, syntenin-1, CD9, calnexin, and actinin-4.

**Figure 3 pharmaceutics-15-00718-f003:**
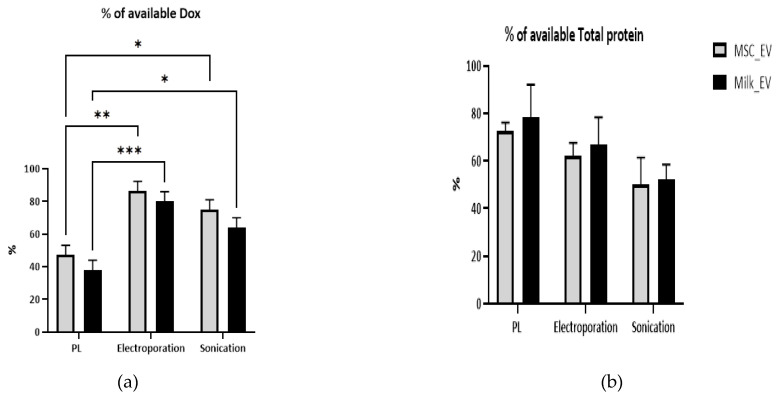
(**a**) Percentage of Dox that was made available and that was apparently loaded into the MSC EVs or milk Evs following passive loading (PL), active loading by electroporation (ALE), or active loading by sonication (ALS). (**b**) Percentage of total protein remaining post-loading by PL, ALE, and ALS, where the total protein was used as a surrogate for Evs quantities. Results are mean from *n* > 3 ± SEM biological repeat experiments. A 2-way Anova analysis was performed; * represents *p* value < 0.05, ** *p* < 0.01, and *** *p* < 0.001.

**Figure 4 pharmaceutics-15-00718-f004:**
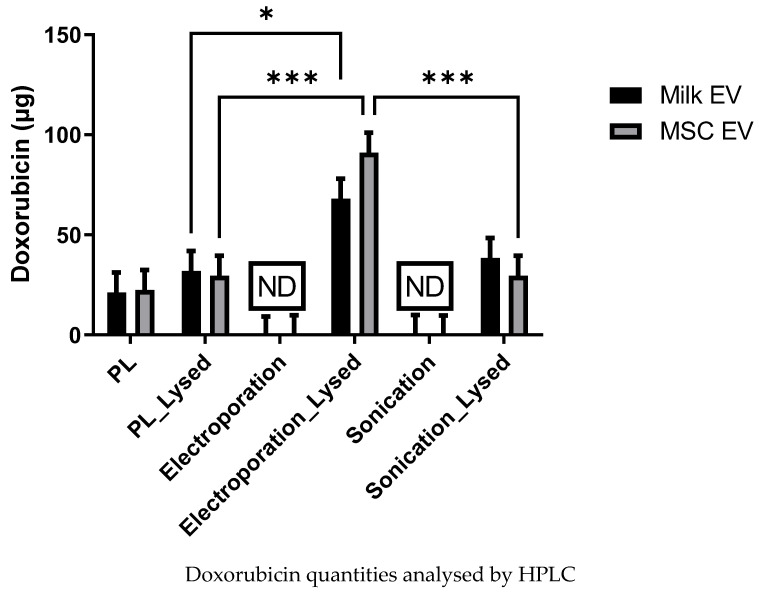
Dox quantities detected for non-lysed compared to lysed passive loaded (PL), electroporation loaded, and sonicated loaded MSC EVs and milk EVs. Results are mean from *n* > 3 ± SEM biological repeat experiments. Statistical analysis was done through 2-way Anova. * represents *p* value < 0.05, and *** *p* < 0.001. ND = not detected.

**Figure 5 pharmaceutics-15-00718-f005:**
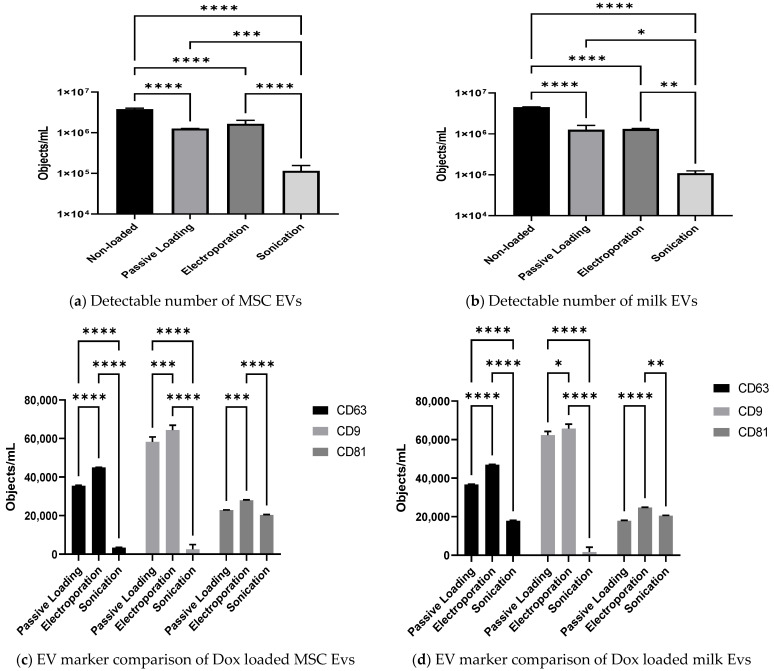
MSC EVs and milk EVs loaded with Dox using passive and active loading approaches were analysed by IFCM to investigate the effect of loading on total detectable EV-like objects and representative EVs surface proteins. Total detectable number of EV-like objects/mL in non-loaded versus Dox-loaded (**a**) MSC EVs and (**b**) milk EVs are presented. Further, the results are presented as a comparison of the number of CD63+, CD9+, and CD81+ objects/EVs in samples in passive loaded and active loaded (**c**) MSC EVs and (**d**) milk EVs. Results are mean from *n* > 3 ± SEM biological repeat experiments. Statistical analysis was done through 2-way Anova; * represents *p* value < 0.05, ** *p* < 0.01, *** *p* < 0.001 and **** *p* < 0.0001.

## Data Availability

Not applicable.

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
