# Peer review of "Doxorubicin Loading into Milk and Mesenchymal Stem Cells’ Extracellular Vesicles as Drug Delivery Vehicles"

_pharmaceutics, 2023, doi:10.3390/pharmaceutics15030718_

Round 1

Reviewer 1 Report

Mukhopadhya et.al have investigated three drug loading techniques on MSC and milk derived EVs. The manuscript is written well, and experiments are performed with proper controls. Findings from this publication will help in developing new therapeutic avenues using EVs as drug loading vehicles. However, few experiments are required assist claims made by the authors before publication. 

1.  Authors claim to have removed casein aggregates from their samples, can you please perform Western Blot on Casein to show the purity. 

2. Authors have successfully incorporated Dox in to the EVs, can the authors show any functional assays (uptake assay or apoptosis assay) on receipt cells showing that these vesicles are still functional after undergoing electroporation and sonication?

Author Response

Reviewer 1 Comments and Suggestions for Authors

Mukhopadhya et.al have investigated three drug loading techniques on MSC and milk derived EVs. The manuscript is written well, and experiments are performed with proper controls. Findings from this publication will help in developing new therapeutic avenues using EVs as drug loading vehicles. However, few experiments are required assist claims made by the authors before publication.

Reply: We thank the Reviewer for the kind words.

We will try to provide clarifications below to support any queries raised. 

  1. Authors claim to have removed casein aggregates from their samples, can you please perform Western Blot on Casein to show the purity. 

Reply: We thank the Reviewer for the comment. To ensure that IP treatment removed major milk proteins, including caseins, untreated milk sample was used as control. In the Figure below (now added as Supplementary Figure 1), presenting total protein separation by polyacrylamide gel electrophoresis of untreated milk sample vs IP treated milk sample, complete removal of caseins in the IP treated samples was observed.

Fig: Total protein separation by polyacrylamide gel electrophoresis indicates efficient removal of casein proteins in IP treated milk samples compared to untreated milk samples.

This has now been mentioned in line 203-205 (Section 3.1) in main manuscript as:

As presented in Supplementary Figure 1, the complete removal of contaminant milk protein, casein, was confirmed by comparing the same with untreated milk EVs samples by polyacrylamide gel electrophoresis.

  1. Authors have successfully incorporated Dox in to the EVs, can the authors show any functional assays (uptake assay or apoptosis assay) on receipt cells showing that these vesicles are still functional after undergoing electroporation and sonication?

Reply: We would like to thank the Reviewer for this suggestion. Within the lab we are currently in the process of designing and performing assays on recipient cells confirming the functional viability of EVs post loading processes, comparing different sources. However, at present, we believe this falls outside the aim and conclusion of this manuscript, which proves that not only MSCs but also milk is a scalable source of EVs and that EVs from both sources are equally suitable. While we do see that electroporation is an efficient method to load doxorubicin in EVs, we can only conclude that both EV sources should be explored to identify an efficient drug delivery vehicle.

A few sentences outlining limitations and future directions of this study have now been added to the Conclusion Section of the manuscript (lines 394-400).

Then comparing MSC EVs to milk EVs, no substantial differences in Dox loading capacity were observed between electroporated MSC EVs and milk EVs, indicating that both are equally suitable. Thus, the decision to select one over the other may depend on the further future experiments, that will take into consideration that MSC EVs have been shown to have some natural anti-inflammatory/immune modulating characteristics that may be of benefit to a given application, in addition to its cargo/loaded therapeutic. On the other hand, milk is cheap, easily accessible, easily scalable, and very rich source of EVs and so may be more favourable in other circumstances, e.g. for oral delivery.

Reviewer 2 Report

This article compared two different exosomes from two different sources. Figure 1 illustrate very clear. But I still have some questions.

First, because this article focuses on compared different exosomes, I think liposomes didn’t conform to the subject (introduction: first paragraph).

Second, about the doxorubicin loading into EVs part, how would you remove the extra medicine or doxorubicin didn’t loading into EVs. Are 250 µg of doxorubicin can perfect loading in the 125 µg of EVs?

Third, about figure 3, could you please added another milk EV group that didn’t remove predominant milk protein. In order to prove your hypothesis.

Fourth, about figure 4. This figure was still explaining the loading condition, So I think it can be remove into the figure 3.

Fifth, about figure 5a,b, Would you please provide other publication to support the results (sonication being a harsh approach that leads to the loss of intact EVs). Because I think electroporation is easier to broken the EV compared sonication.

Sixth, could you please add other experiment to explain different change about the EVs surface markers. This article just shown a phenomenon, but did not explain why it will change and if it is change what will happen.

Author Response

Reviewer 2 Comments and Suggestions for Authors

This article compared two different exosomes from two different sources. Figure 1 illustrate very clear. But I still have some questions.

Reply: Thank you and we are glad to be provided an opportunity to address your questions. 

First, because this article focuses on compared different exosomes, I think liposomes didn’t conform to the subject (introduction: first paragraph).

Reply: Indeed, we agree to this comment and edited this sentence accordingly. The lines 41-45 have now been edited accordingly.  

Second, about the doxorubicin loading into EVs part, how would you remove the extra medicine or doxorubicin didn’t loading into EVs. Are 250 µg of doxorubicin can perfect loading in the 125 µg of EVs?

Reply: We thank the Reviewer for pointing this out to us. A washing step was included after the loading step. The loaded EVs were washed using a 300 kDa filter (Nanosep, Pall Biotech, Ireland). This information has been included in the materials and methods section as below:

Lines 171-172: Following the loading step, unbound Dox was removed by washing. The loaded EVs were washed using a 300 kDa filter (Nanosep, Pall Biotech, Ireland).

Are 250 µg of doxorubicin can perfect loading in the 125 µg of EVs?

Reply: Yes.

Third, about Figure 3, could you please added another milk EV group that didn’t remove predominant milk protein. In order to prove your hypothesis.

Reply: Thanks. As a control sample to ensure milk proteins, especially caseins, were efficiently removed by isoelectric precipitation, an untreated milk sample was parallelly processed. Following EVs separation, the total proteins of the 2 samples were separated via polyacrylamide gel electrophoresis. As presented in the Figure below, a clear removal of major milk protein, casein, was observed, confirming that IP removes casein that may act as a contaminant.

Fig: Total protein separation by polyacrylamide gel electrophoresis indicates efficient removal of casein proteins in IP treated milk samples compared to untreated milk samples.

This has been now mentioned in line 203-205 (Section 3.1) in main manuscript as:

As presented in Supplementary Figure 1, the complete removal of contaminant milk protein, casein, was confirmed by comparing untreated milk EV samples by polyacrylamide gel electrophoresis.

Fourth, about figure 4. This figure was still explaining the loading condition, So I think it can be remove into the figure 3.

Reply: We would like to clarify here that the purpose of presenting Figure 3 and Figure 4 separately is because they present 2 different techniques of evaluation of doxorubicin loading.

Specifically, Figure 3 represents the percentage of doxorubicin and related percentage of EV proteins available after the loading process. This was performed by measurement of the fluorescence of doxorubicin, as well as the protein concentration using Bradford protein quantification.

In Figure 4, the quantity of doxorubicin that is loaded in the EVs was quantified by high performance liquid chromatography (HPLC). Hence, we feel that although both Figures represent loading of doxorubicin in EVs, they involve 2 different techniques of quantification and are best illustrated as 2 separate Figures.

However, if the Reviewer strongly believes that it is necessary to join as one Figure, we will be happy to do so.

Fifth, about figure 5a,b, Would you please provide other publication to support the results (sonication being a harsh approach that leads to the loss of intact EVs). Because I think electroporation is easier to broken the EV compared sonication.

Reply: Thank you for bringing this topic up. In fact, as per the literature search, there is no clear consensus on the most suitable drug loading method into EVs, thus far. Additionally consideration should be given to the characterisation methods used for both EVs and doxorubicin. In order to overcome these inconsistencies, we designed this study as reported because we wanted to compare EVs from 2 different sources that undergoes same loading method and is characterised on similar platforms to generate comparable and reproducible results. We chose 3 loading methods based on a literature search encompassing a mild technique (incubation/passive loading) and 2 harsher techniques/active loading (electroporation and sonication).

Overall, the characterisation of loaded EVs did not show any difference between the 2 EV sources. However, we did observe differences in loading capacities and differences in the surface proteins of EVs post-sonication. Sonication has been known to be a harsh technique that can cause irreversible disruption to the cellular membrane (Srivastava et al., 2020). This is a plausible explanation as yo why we see fewer detectable EVs following sonication (Fig 5a,b). A publication by Zubair Ahmed et al., (2021) reported that ‘low-power sonication’ was able to alter MSC EVs membrane integrity. Hence, to identify an efficient EV loading technique, it is of utmost importance to implement standardised reporting of characterisation methods.  

  • Srivastava, Akhil, et al. "Progress in extracellular vesicle biology and their application in cancer medicine." Wiley Interdisciplinary Reviews: Nanomedicine and Nanobiotechnology4 (2020): e1621.
  • Nizamudeen, Zubair Ahmed, et al. "Low-power sonication can alter extracellular vesicle size and properties." Cells9 (2021): 2413.

These have now been cited in the main manuscript as below:

Lines 340-344: Sonication has been known to be a harsh technique that can cause irreversible disruption to the cellular membrane (33), this may explain lower number of detectable EVs following sonication (Fig 5a,b). Additionally, Zubair Ahmed et al. (34) reported that ‘low-power sonication’ altered MSC EVs membrane integrity.

Sixth, could you please add other experiment to explain different change about the EVs surface markers. This article just shown a phenomenon, but did not explain why it will change and if it is change what will happen.

Reply: We thank the Reviewer for the suggestion. Indeed, the research group has been working towards designing and performing downstream experiments to prove that the sources of EVs, following EV loading, are functional in in vitro and in vivo systems. However, for the current manuscript, we would like to focus on the message that there are no major detectable differences in the EV sources and indeed both MSC and milk are 2 scalable sources of EVs and should be further explored as drug delivery vehicles. In future experiments when additional research funds are available, we would like to consider how efficiently the EVs deliver active drug compounds to targeted cells, as well as what the effects are on the functionality of the EVs after processing by relatively harsh techniques such as electroporation and sonication. 

A few sentences relating to limitations and potential future directions of the study have now been added to the Conclusion Section of the manuscript (lines 394-400)

Then comparing MSC EVs to milk EVs, no substantial differences in Dox loading capacity were observed between electroporated MSC EVs and milk EVs, indicating that both are equally suitable. Thus, the decision to select one over the other may depend on the further future experiments, that will take into consideration that MSC EVs have been shown to have some natural anti-inflammatory/immune modulating characteristics that may be of benefit to a given application, complementing a loaded therapeutic. On the other hand, milk is a cheap, easily accessible, easily scalable very rich source of EVs the biological functions of which are less explored but have great potential for oral delivery.

Round 2

Reviewer 1 Report

Authors have provided justifications for the concerns that were there in the first draft of the manuscript. The manuscript can now be accepted in the current form.